# Effect of Angiotensin II on Bone Erosion and Systemic Bone Loss in Mice with Tumor Necrosis Factor-Mediated Arthritis

**DOI:** 10.3390/ijms21114145

**Published:** 2020-06-10

**Authors:** Takahiko Akagi, Tomoyuki Mukai, Takafumi Mito, Kyoko Kawahara, Shoko Tsuji, Shunichi Fujita, Haruhito A. Uchida, Yoshitaka Morita

**Affiliations:** 1Department of Rheumatology, Kawasaki Medical School, Kurashiki, Okayama 701-0192, Japan; akagitahiko@gmail.com (T.A.); mito.tac@gmail.com (T.M.); kyoko.k0925@gmail.com (K.K.); shoko7.05.13@gmail.com (S.T.); shunic117@gmail.com (S.F.); morita@med.kawasaki-m.ac.jp (Y.M.); 2Department of Chronic Kidney Disease and Cardiovascular Disease, Okayama University Graduate School of Medicine, Dentistry and Pharmaceutical Sciences, Okayama 700-0914, Japan; hauchida@okayama-u.ac.jp

**Keywords:** angiotensin II, arthritis, bone erosion, inflammation, tumor necrosis factor, renin-angiotensin system, angiotensin II type 1 receptor

## Abstract

Angiotensin II (Ang II) is the main effector peptide of the renin-angiotensin system (RAS), which regulates the cardiovascular system. The RAS is reportedly also involved in bone metabolism. The upregulation of RAS components has been shown in arthritic synovial tissues, suggesting the potential involvement of Ang II in arthritis. Accordingly, in the present study, we investigated the role of Ang II in bone erosion and systemic bone loss in arthritis. Ang II was infused by osmotic pumps in tumor necrosis factor-transgenic (TNFtg) mice. Ang II infusion did not significantly affect the severity of clinical and histological inflammation, whereas bone erosion in the inflamed joints was significantly augmented. Ang II administration did not affect the bone mass of the tibia or vertebra. To suppress endogenous Ang II, Ang II type 1 receptor (AT1R)-deficient mice were crossed with TNFtg mice. Genetic deletion of AT1R did not significantly affect inflammation, bone erosion, or systemic bone loss. These results suggest that excessive systemic activation of the RAS can be a risk factor for progressive joint destruction. Our findings indicate an important implication for the pathogenesis of inflammatory bone destruction and for the clinical use of RAS inhibitors in patients with rheumatoid arthritis.

## 1. Introduction

Rheumatoid arthritis is a chronic inflammatory disorder that can cause painful swelling and bone erosion in the inflamed joints [1]. The accumulation of joint damage results in long-lasting pain and deformity of the affected joints [2]. Persistent systemic inflammation in rheumatoid arthritis can also cause tissue damage in organs such as the lungs, heart, eyes, and bone [3]. Increased inflammatory cytokines affect bone metabolism throughout the body and decrease bone mass and strength, leading to increased risks of osteoporosis and fracture [4]. Joint deformities impair activities of daily life and thus exacerbate osteoporosis in patients with rheumatoid arthritis. This highlights the importance of resolving joint damage and systemic bone loss issues in these patients.

Angiotensin II (Ang II) is the main effector peptide of the renin-angiotensin system (RAS), which causes vasoconstriction and an increase in blood pressure [5,6]. The RAS is one of the most important systems in cardiovascular control [5]. Ang II is generated from its precursor angiotensinogen through serial enzymatic processes mediated by renin and angiotensin-converting enzyme (ACE). Ang II is converted mainly in the kidneys and lungs. Recent studies have shown that Ang II is generated locally in various tissues such as the brain, vascular wall, and reproductive tract [5,7]. Locally produced Ang II plays a role in many physiological and pathological processes such as hypertension, inflammation, tissue fibrosis, and oxidative stress [8], and it fulfills a key function in bone metabolism [6,9]. Animal studies have revealed that Ang II and Ang II type1 receptor (AT1R) are expressed locally within bone tissue and regulate bone mass [6,9,10]. Additionally, studies using mouse models have indicated that the local RAS is involved in the pathogenesis of several bone diseases, such as postmenopausal [11], age-related [12], and glucocorticoid-induced osteoporosis [13].

Components of the RAS are expressed at elevated levels in the synovium of patients with rheumatoid arthritis compared to those in the non-inflamed synovium [9,14,15]. However, the role of the RAS in the mechanisms underlying bone erosion and inflammation-mediated bone loss in arthritic conditions remains unclear. In the present study, we investigated the in vivo effects of Ang II on bone erosion and systemic bone loss in a tumor necrosis factor (TNF)-induced arthritis model. We also aimed to assess whether the deletion of AT1R ameliorates the bone erosion and systemic bone loss caused by arthritis.

## 2. Results

### 2.1. Increased AT1R Expression in the Joint Tissue of Tumor Necrosis Factor-Transgenic (TNFtg) Mice

As a first step to explore the effect of Ang II on arthritis, the expression of *Agtr1a*, encoding AT1R, in the joints was assessed by quantitative polymerase chain reaction (qPCR). *Agtr1a* mRNA was detected in the joint tissues from both wild-type (WT) and TNFtg mice, and the expression levels were 2.5-fold elevated in the tissues of the TNFtg mice compared with those in the WT mice (Figure 1A). Increased AT1R protein expression was also shown in arthritic joint tissues by Western blotting (Figure A1A). Immunohistochemical stain of the arthritic joints revealed AT1R expression in the tissues, including proliferated synovial cells (Figure A2B).

### 2.2. No Significant Effect of Ang II Administration on the Severity of Inflammatory Cell Infiltration in TNFtg Mice

To assess the effect of Ang II on arthritis, exogenous Ang II (1.44 mg/kg/day) or water (H_2_O) was administered by osmotic pumps to the WT and TNFtg mice for 4 weeks. The treatment with Ang II did not significantly alter body weight (Figure 1B), but did induce hypertension (Figure A2). We monitored the severity of paw swelling in each limb during the experimental period. We found that the TNFtg mice exhibited severe swelling of the paws and that the severity of clinical arthritis was not affected by the Ang II infusion. The arthritis score and number of arthritic limbs at the age of 16 weeks are presented in Figure 1C,D.

To analyze the inflamed joints histologically, we performed hematoxylin and eosin (H&E) and Safranin O staining to determine the inflammatory cell infiltration and cartilage damage. In WT mice, Ang II administration did not cause any detectable histological changes (Figure 1E,F). TNFtg mice exhibited massive inflammatory cell infiltration, and Ang II administration did not affect the severity of inflammation in these mice, which is consistent with the arthritis score results (Figure 1C). Additionally, the severity of cartilage damage, represented by decreased staining of the cartilage matrix, was not affected by the administration of Ang II (Figure 1E,G).

### 2.3. Exacerbation of Bone Erosion by Ang II Administration in TNFtg Mice

We then examined the impact of Ang II on the erosive bone changes of the ankle. Bone erosion around the talus was quantified using micro-computed tomography (CT) and 3D image analysis software. The micro-CT analysis revealed that the destructive bone change was significantly more severe in the Ang II-infused TNFtg mice than in the H_2_O-infused TNFtg mice (Figure 2A). This aggravated bone erosion was revealed by the following quantitative analyses: the bone volume (BV) of the talus, the reduction rate of BV, and the eroded volume per repaired volume (Ev/Rpv) of the talus (Figure 2B–D). Tartrate-resistant acid phosphatase (TRAP)-stained images showed slightly increased osteoclast formation in the joints of the Ang II-infused TNFtg mice compared to those in the H_2_O-infused TNFtg mice (Figure 2E). Quantitative histological analyses also revealed increased bone erosion and slightly enhanced osteoclast formation around the talus (Figure 2F,G). These findings suggest that Ang II, imported to the joints from circulation, served as an osteoclast-activating factor in the arthritic joint, resulting in enhanced bone erosion without affecting the clinical severity of arthritis.

### 2.4. No Detectable Changes in the Trabecular and Cortical Bone Parameters with Ang II Administration

Since both systemic inflammation and excess of Ang II have been reported to decrease the mass of systemic bones [4,11], we examined the bone properties of the tibia and vertebra and determined whether Ang II could synergistically enhance inflammation-mediated bone loss in the Ang II-administered arthritic mice. We assessed the tibia trabecular bone (secondary spongiosa, Figure 3A), the tibia cortical bone (midshaft of the tibia, Figure 3B), and the trabecular bone of the spine (fifth lumbar vertebra, Figure 3C) using micro-CT. The tibia trabecular bone tended to be decreased in the arthritic mice compared to that in the WT mice, although the difference was not statistically significant in the current set of experiments. The bone reduction rates with Ang II infusion were comparable between WT and TNFtg mice at approximately 30% (Figure 3D), indicating that the synergistic effect of inflammation and Ang II on osteopenia was not noticeable. In the tibia cortical bone, the presence of arthritis significantly increased bone loss, but the reduction rates with Ang II infusion were comparable between WT and TNFtg mice (Figure 3E). In the vertebral trabecular bone, the presence of arthritis did not significantly affect the bone volume (Figure 3F). Ang II administration tended to decrease bone volume by approximately 10%, but there was no significant difference in the reduction rates between WT and TNFtg mice (Figure 3F). The other analyzed parameters of the trabecular and cortical bones also indicated no significant effect of Ang II administration on bone properties (Figure A3). Collectively, these findings suggest that both inflammation and Ang II tended to decrease bone mass, but there was no apparent synergistic effect on the osteopenic phenotype.

### 2.5. Effect of AT1R Deficiency on the Severity of Inflammatory Cell Infiltration in TNFtg Mice

Since an excess of exogenous Ang II accelerated inflammatory bone destruction (Figure 2A), we investigated whether endogenous Ang II could play a role in bone destruction using AT1R-deficient arthritic mice generated by crossing TNFtg mice with AT1R-knockout (AT1R^−/−^) mice. AT1R deficiency did not significantly alter body weight (Figure 4A). We found that the severity of clinical arthritis (Figure 4B,C) and the extent of inflammatory cell infiltration (Figure 4D,E) were comparable between TNFtg and TNFtg/AT1R^−/−^ mice. In addition, the severity of cartilage damage was not affected by AT1R deficiency (Figure 4D,F).

### 2.6. Influence of AT1R Depletion on Bone Erosion in TNFtg Mice

We next examined whether the deletion of AT1R could reduce bone destruction in the arthritic mice. Micro-CT analysis of the ankle joints revealed that TNFtg/AT1R^−/−^ mice exhibited the same extent of severe bone loss as TNFtg mice (Figure 5A,B). Additionally, the BV reduction rate and the erosive volume (Ev/Rpv) of the talus in the TNFtg/AT1R^−/−^ mice were comparable to those in the TNFtg mice (Figure 5C,D). These findings indicate that AT1R deficiency did not alleviate the destructive bone changes in the inflammatory joints of the arthritic mice. Histological analyses revealed that the extents of bone erosion and osteoclast formation were comparable between TNFtg and TNFtg/AT1R^−/−^ mice (Figure 5E–G).

### 2.7. Effect of AT1R Deficiency on Bone Properties of the Trabecular and Cortical Bones in TNFtg Mice

AT1R^−/−^ mice were previously reported to exhibit an increased trabecular BV and increased trabecular number and connectivity [16]. To examine the effect of AT1R deficiency on the bone volume of systemic bones in the arthritic condition, we analyzed the bone properties of the tibia and vertebra of the TNFtg arthritic mice using micro-CT (Figure 6A–C). The TNFtg mice exhibited a significant reduction in BV/TV of the tibia and the AT1R deficiency modestly alleviated the bone loss caused by arthritis, even though the difference between TNFtg and TNFtg/AT1R^−/−^ mice was not statistically significant (Figure 6D). A similar insignificant tendency was observed in the vertebral trabecular bone (Figure 6F). In the tibia cortical bone, AT1R deficiency did not show any protective effect on bone loss (Figure 6E). The other analyzed parameters of the trabecular and cortical bones also indicated no significant effect of AT1R deficiency on bone properties (Figure A4). These findings suggest that the inhibition of endogenous Ang II has a limited protective effect on bone loss in arthritic mice.

## 3. Discussion

In this study, we sought to clarify the impact of excessive Ang II and inhibition of the endogenous RAS on bone erosion and systemic bone loss in a TNF-mediated arthritic condition. We found that the administration of Ang II enhanced destructive bone changes in inflammatory joints without affecting the severity of inflammation. There was no noticeable synergistic effect of Ang II administration and inflammation on osteopenia of the tibia and vertebra in mice. Further, we found that AT1R deficiency had a minimal protective effect on bone erosion and systemic bone loss in the arthritis model.

Interestingly, we observed that the administration of Ang II aggravated joint destruction in the arthritic mice. Ang II has been reported to enhance systemic bone loss in murine osteoporosis models [11,17]. Ang II induces RANKL expression in osteoblasts and subsequently enhances osteoclastogenesis, resulting in systemic bone loss [6,11]. However, no previous studies have explored the role of the RAS in the development of bone erosion in an arthritis model. In rheumatoid arthritis, inflammatory cytokines such as TNF increase RANKL expression in synoviocytes and subsequently promote osteoclastic differentiation and activation, resulting in erosive bone changes in joints [2]. Our results demonstrate that excessive Ang II could exacerbate the TNF-induced inflammatory joint destruction associated with increased osteoclast formation.

The current study has important clinical implications for the management of rheumatoid arthritis. Our findings suggest that in patients in whom the local effect of Ang II is upregulated via increased imported Ang II from circulation, joint destruction can be promoted as a consequence of systemic RAS activation. Systemic activation of the RAS can be observed in several pathological conditions, such as renal artery stenosis, congestive heart failure, cardiac hypertrophy, chronic kidney disease, and obesity [18,19]. Such pathological conditions could be risk factors for progressive joint destruction in inflammatory arthropathies.

Although Ang II appeared to promote bone erosion in inflamed joints, its effects on systemic bones, represented by the tibia and vertebra, were found to be very limited. There are several possible explanations for this. Firstly, in the arthritic joints of mice, other inflammatory cytokines such as IL-1 and IL-6 are highly produced [2,20]. In addition to TNF, these other osteoclast-activating factors might play important synergistic roles in the Ang II-promoted bone erosion in joints. Secondly, the expression of AT1R was significantly increased in the arthritic joints (Figure 1A). This could contribute to hyper-responsiveness to Ang II, resulting in increased osteoclastic bone destruction in the joints. Thirdly, the exposure period to Ang II (4 weeks in this study) might be too short for this effector to exert an osteopenic effect on systemic bones. Indeed, a previous study showed a significant osteopenic effect of excessive RAS activation in 6-month-old Tsukuba hypertensive mice that were continuously exposed to excessive Ang II via transgenes encoding human renin and human angiotensinogen [17].

Since the expression of AT1R was increased in the arthritic joints of the TNFtg mice (Figure 1A), we assumed that AT1R deficiency would ameliorate bone erosion in this arthritis model. Contrary to our expectation, AT1R deficiency did not significantly improve the erosive bone changes in the TNFtg mice. These data indicate a limited role of the local RAS during the process of joint destruction in this arthritic model. The RAS might modulate bone mass only under pathological conditions with excessive systemic activation. Analyses of AT1R-deficient arthritic mice with excessive Ang II would be needed to verify this concept.

The limitation of our study is that the precise mechanisms through which Ang II enhances bone erosion are unclear. We have tested the effect of Ang II on osteoclast differentiation in murine primary bone marrow-derived macrophage cultures. Ang II stimulation did not promote osteoclast formation in the mono-culture of bone marrow-derived macrophage (Figure A5A,B), whereas Ang II enhanced osteoclast formation in the co-culture system with osteoblasts (Figure A5C,D). These data suggest that Ang II promotes osteoclast formation indirectly via stromal cells. In support of this notion, Ang II has been previously reported to induce RANKL expression in stromal cells [17]. Various cells can express RANKL in arthritic joints, synovial cells, osteoblasts, or osteocytes, which might be attributed to the Ang II-mediated bone erosion. Other possibilities are that Ang II regulates angiogenesis in arthritic joints or that Ang II modulates cellular functions via the Ang II type 2 receptor, which reportedly regulates inflammation in the arthritic synovium [21]. Further research will be required to clarify the underlying mechanisms.

Another possible limitation of this study is the relatively small sample sizes which may have insufficient statistical power to detect a small difference in some comparisons. For instance, a statistically significant difference was not detected in the trabecular BV/TV of the tibia between H_2_O-treated WT (*n* = 6) and H_2_O-treated TNFtg (*n* = 4) mice (Figure 3D), although there is a statistically significant difference between WT (*n* = 9) and TNFtg (*n* = 12) mice in the AT1R^−/−^ strain (Figure 6D). Post-hoc power analyses have shown that a larger sample size would be needed to detect a substantial difference in Figure 3D. Therefore, future studies with larger sample sizes would be necessary to detect a small but significant difference.

We previously reported that the RAS is involved in vascular damage and that AT1R blockers have potent vascular protective effects in an arthritis model [22]. Therefore, in patients with rheumatoid arthritis complicated by RAS-dependent hypertension, the blockade of the RAS might be beneficial not only to reduce blood pressure and vascular damage but also to prevent bone erosion.

In conclusion, this study provides novel insights into the pathophysiological function of Ang II in the regulation of inflammatory bone destruction. In patients with rheumatoid arthritis, the systemically activated RAS in concurrent pathological conditions could be involved in the progression of joint destruction in conjunction with increased local expression of AT1R. The effects of pharmacological inhibition of the Ang II-mediated pathway on bone erosion remain unclear but warrant further clinical examination.

## 4. Materials and Methods

### 4.1. Mice

Human TNFtg mice (C57BL/6 background) were obtained (#1006; Taconic Biosciences, Hudson, NY, USA). The TNFtg heterozygous mice spontaneously develop arthritis on the fore and hind paws at approximately 8 weeks of age, and arthritis progresses with age [23]. AT1R-knockout mice (AT1R^−/−^; C57BL/6 background) were obtained (#002682; The Jackson Laboratory, Bar Harbor, ME, USA) [24] and crossed with TNFtg mice to generate AT1R-deficient arthritic mice. Age- and sex-matched littermates were used as control mice. All mutant mice were maintained in the animal facility of Kawasaki Medical School (Okayama, Japan) and were housed in a group (2–5 mice per cage) and maintained at 22 °C under 12 h light/12 h dark cycles with free access to water and standard laboratory food. All animal experiments were approved by the Institutional Safety Committee for Recombinant DNA Experiments (Nos. 14–40, 14–41, and 19–27, which are approved on 3/13/2015, 3/13/2015, and 10/17/2019, respectively) and the Institutional Animal Care and Use Committee of Kawasaki Medical School (Nos. 17–129, 18–057, and 18–130, which are approved on 2/1/2018, 4/1/2018, and 2/1/2019, respectively). All experimental procedures were conducted in accordance with institutional and NIH guidelines for the humane use of animals.

### 4.2. Ang II Infusion Model

Twelve-week-old WT and TNFtg male mice were randomly divided into two groups that were infused with either water (H_2_O) or Ang II, which was dissolved in H_2_O. Ang II was administered by osmotic pumps to WT (*n* = 6) and TNFtg mice (*n* = 7) from 12 to 16 weeks of age. H_2_O was administered by osmotic pumps to WT (*n* = 6) and TNFtg mice (*n* = 4) as controls. The mice were anesthetized, and an osmotic pump containing 100 μL of either H_2_O or Ang II (Sigma-Aldrich, St. Louis, MO, USA) was implanted subcutaneously as previously described [22,25]. Ang II was continuously infused at a dose of 1.44 mg/kg/day from 12 to 16 weeks of age. Arterial blood pressure was measured by the tail-cuff method with a pulse transducer (BP98-A; Softron, Tokyo, Japan), as reported [26]. Mice were monitored for signs of arthritis in a blinded manner, and each limb was individually scored on a scale of 0–4. Scores were assigned based on the extent of erythema or swelling present in each limb, assigning a maximum score of 16 per mouse, as described previously [27,28]. Mice were monitored until the age of 16 weeks, and then serum, hind limb, and spine (the fifth lumbar vertebra) samples were collected.

### 4.3. Micro-Computed Tomography (CT) Analysis

Bone samples were fixed in 4% paraformaldehyde (PFA) in phosphate-buffered saline for 2 days, and PFA-fixed bone samples were immersed in 70% ethanol. Three-dimensional microarchitecture of the talus, tibia, and spine was evaluated by using a micro-CT system (Ele Scan mini; Nittetsu Elex, Tokyo, Japan) with an X-ray energy of 45 kVP (145 μA), as described previously [29,30]. The voxel resolution of all bone images was 15 μm. The bone properties of the tibia and the fifth lumbar vertebra, and bone erosion of the ankle (talus bones) were analyzed using analysis software (TRI/3D-BON; Ratoc System Engineering Co. Ltd., Tokyo, Japan). The analyzed region of the tibia trabecular bone comprised 67 slices of secondary spongiosa adjacent to the primary spongiosa (starting 0.5 mm from the distal border of the growth plate), that of the vertebra comprised the entire fifth lumbar vertebral body area (approximately 140 slices), and that of the tibia cortical bone comprised 33 slices of the midshaft (1 mm proximal region from the tibiofibular junction). The micro-CT parameters of the tibia and spine were described according to international guidelines [31]. The talus bones were evaluated using BV and Ev/Rpv for quantitative measurements of bone erosion [32]. Ev/Rpv on the whole talus was calculated automatically with the software according to the software program (TRI/3D-BON). We set the concave surface search range up to 0.15 mm, and the absorption surface extraction radius of curvature was 960 μm or less as described previously [33].

### 4.4. Histological Analysis

The hind limbs were decalcified in 10% EDTA (pH 7.2) at 4 °C for 4 weeks and subsequently embedded in paraffin. Sections (3 µm) were stained with hematoxylin and eosin (H&E) and Safranin O. The severity of inflammation and cartilage damage around the talus bone was scored on a scale of 0–4 under blinded conditions as described previously [27,28]. TRAP staining was performed to visualize osteoclast formation, and the sections were counterstained with methyl green. Histological analyses were performed using a BZ-X analyzer (Keyence, Osaka, Japan). The eroded surface per bone surface (ES/BS) and the number of osteoclasts per bone surface (N.Oc/BS) around the taluses were determined.

### 4.5. Real-Time Quantitative Polymerase Chain Reaction (qPCR)

qPCR was performed as described previously [30,34]. Total RNA was extracted from the right ankle joint using RNAiso Plus (Takara Bio, Shiga, Japan) and solubilized in ribonuclease (RNase)-free water. Complementary DNA (cDNA) was synthesized using the Prime Script RT reagent Kit (Takara Bio). qPCR reactions were performed using SYBR Green PCR Master Mix (Takara Bio) with the StepOnePlus Real-Time PCR System (Thermo Fisher Scientific, Waltham, MA, USA). Gene expression levels relative to *Gapdh* were calculated by the ΔΔCt method and normalized to control samples obtained from the WT mice. The qPCR analysis used the following primers: 5′-taccagctctgcggctct-3′ and 5′-gccagccattttataccaatct-3′ for *Agtr1a* (AT1R); 5′-atcaagaaggtggtgaagca-3′ and 5′-gacaacctggtcctcagtgt-3′ for *Gapdh*. All qPCR reactions yielded products with single peak dissociation curves.

### 4.6. Statistical Analysis

All values are given as the mean ± standard error of the mean (SEM). A two-tailed unpaired Student’s *t*-test was used to compare two groups, and a one-way analysis of variance (ANOVA) followed by Tukey’s post-hoc test was used to compare three or more groups by using GraphPad Prism 5 (GraphPad Software, San Diego, CA, USA). *p* values lower than 0.05 were considered statistically significant.

### 4.7. Supplementary Methods

Additional Appendix A can be found online in the Appendix A tab for this article.

## Figures and Tables

**Figure 1 ijms-21-04145-f001:**
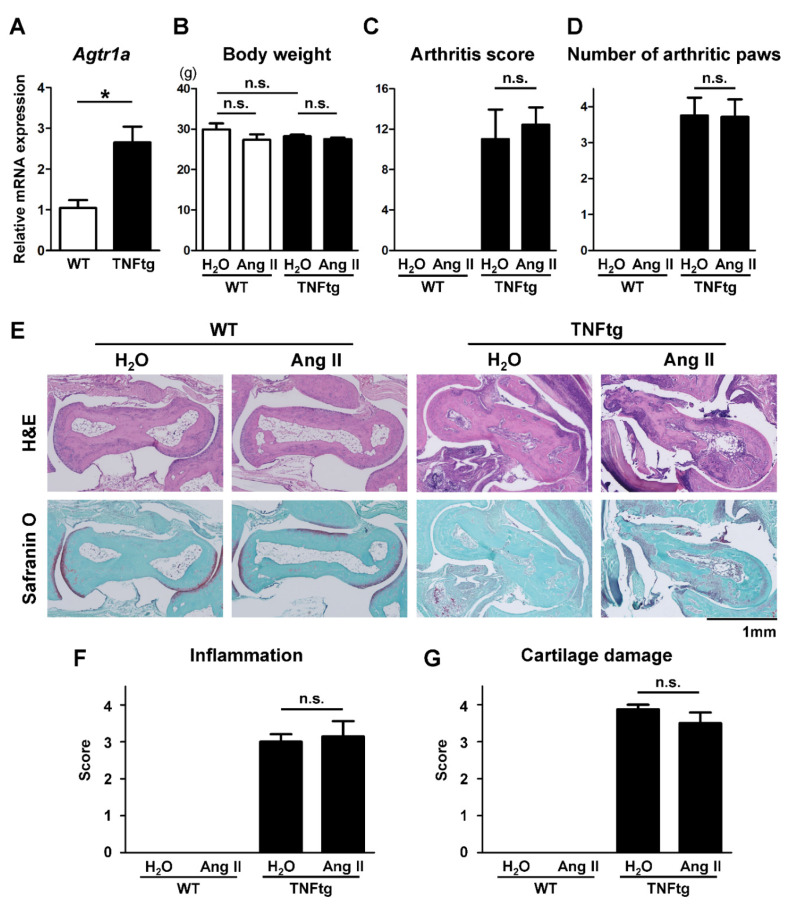
No significant effect of angiotensin II (Ang II) administration on the severity of inflammatory cell infiltration and cartilage damage in tumor necrosis factor-transgenic (TNFtg) mice. Ang II was administered by osmotic pumps to wild-type (WT; *n* = 6) and TNFtg (*n* = 7) male mice from 12 to 16 weeks of age. Water (H_2_O) was administered by osmotic pumps to WT (*n* = 6) and TNFtg (*n* = 4) male mice as controls. (**A**) Quantitative real-time PCR analysis. *Agtr1a* mRNA expression levels in the right ankle joint tissue were determined. (**B**) Body weight at the age of 16 weeks. (**C**) Arthritis scores at the age of 16 weeks. (**D**) The numbers of arthritic paws with an arthritis score of 2 or higher. (**E**) Representative images of stained sections around the talus bones of indicated mice; sections were stained with hematoxylin and eosin (H&E) and Safranin O; original magnification ×40. (**F**,**G**) Histological scores of inflammation (**F**) and cartilage damage (**G**). Values are the mean ± SEM. n.s., not significant. *, *p* < 0.05.

**Figure 2 ijms-21-04145-f002:**
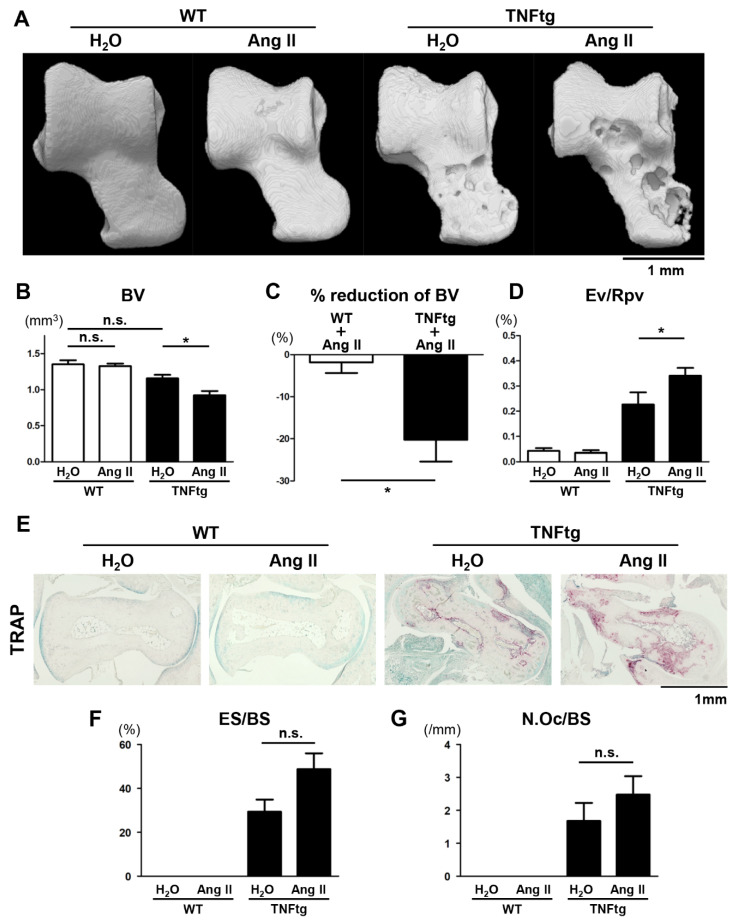
Exacerbation of bone erosion by angiotensin II (Ang II) administration in tumor necrosis factor-transgenic (TNFtg) mice. Ang II was administered by osmotic pumps to WT (*n* = 6) and TNFtg (*n* = 7) male mice from 12 to 16 weeks of age. Water (H_2_O) was administered by osmotic pumps to WT (*n* = 6) and TNFtg (*n* = 4) male mice as controls. (**A**) Representative 3D micro-computed tomography (CT) images of the talus. (**B**) Bone volume (BV) of the talus. (**C**) Rates of reduction in BV of Ang II-infused WT and TNFtg mice relative to H_2_O-infused control mice of each genotype. (**D**) Eroded volume per repaired volume (Ev/Rpv) of the talus. (**E**) Representative images of tartrate-resistant acid phosphatase (TRAP) staining around the talus bones of indicated mice; original magnification ×40. (**F**) Eroded surface per bone surface (ES/BS) around the taluses. (**G**) The number of osteoclasts per bone surface (N.Oc/BS) around the taluses. Values are the mean ± SEM. n.s., not significant. *, *p* < 0.05.

**Figure 3 ijms-21-04145-f003:**
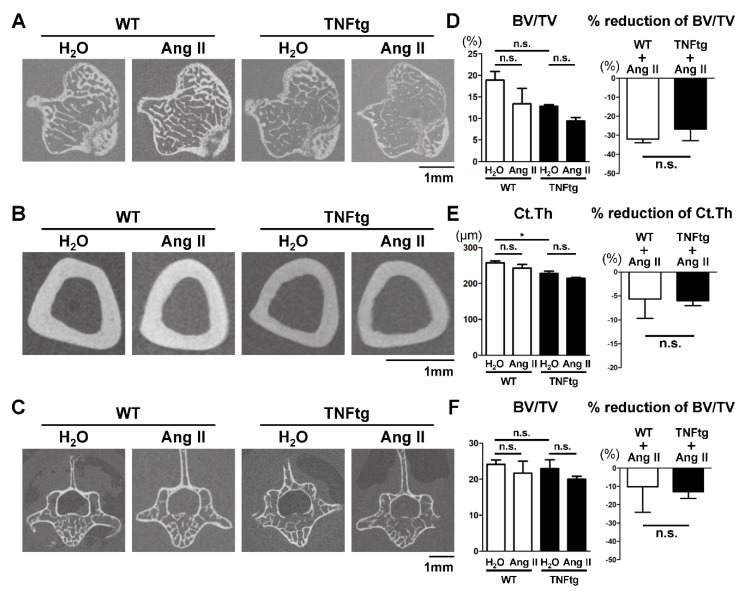
No detectable changes in the trabecular and cortical bone parameters with Ang II administration. Angiotensin II (Ang II) was administered by osmotic pumps to WT (*n* = 6) and tumor necrosis factor-transgenic (TNFtg; *n* = 7) male mice from 12 to 16 weeks old. Water (H_2_O) was administered by osmotic pumps to WT (*n* = 6) and TNFtg (*n* = 4) male mice as controls. (**A**–**C**) Representative 2D micro-CT images of the tibia trabecular bone (**A**), the tibia cortical bone (**B**), and the trabecular bone of the spine (the fifth lumbar vertebra) (**C**). (**D**) Bone volume per total volume (BV/TV) and reduction rate of the tibia trabecular bone. (**E**) Cortical thickness (Ct.Th) and reduction rate of the tibia midshaft. (**F**) Bone volume per total volume (BV/TV) and reduction rate of the fifth lumbar vertebral trabecular bone. Values are the mean ± SEM. n.s., not significant. *, *p* < 0.05.

**Figure 4 ijms-21-04145-f004:**
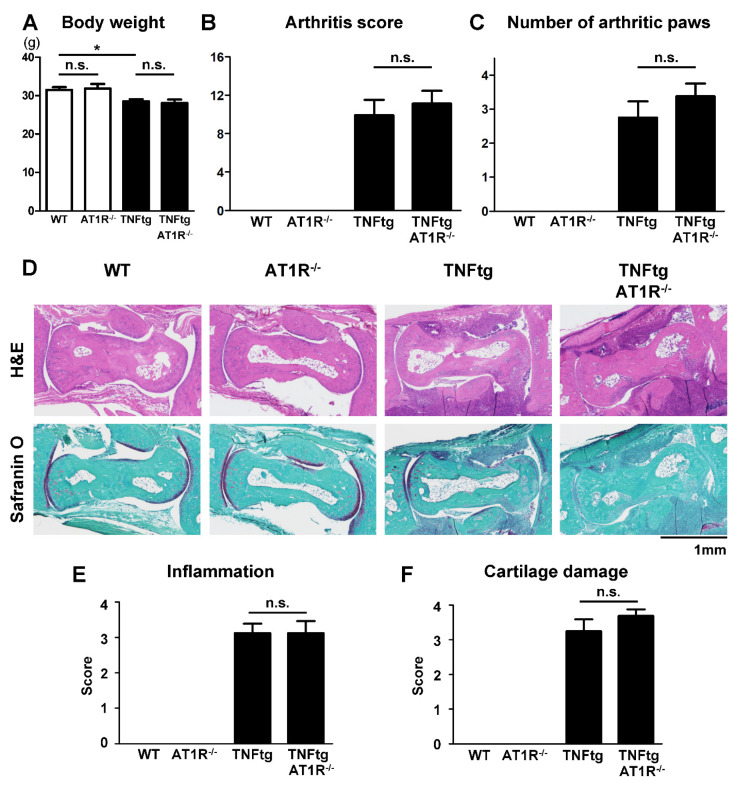
Effect of angiotensin II type 1 receptor (AT1R) deficiency on the severity of inflammatory cell infiltration in tumor necrosis factor-transgenic (TNFtg) mice. TNFtg mice were crossed with AT1R-deficient (AT1R^−/−^) mice. WT (*n* = 9), AT1R^−/−^ (*n* = 7), TNFtg (*n* = 12), and TNFtg AT1R^−/−^ (*n* = 8) male mice were analyzed at the age of 16 weeks. (**A**) Body weight at the age of 16 weeks. (**B**) Arthritis score at the age of 16 weeks. (**C**) The number of arthritic paws with an arthritis score of 2 or higher. (**D**) Representative images of stained sections around the talus bones of indicated mice; sections were stained with hematoxylin and eosin (H&E) and Safranin O; original magnification ×40. (**E**,**F**) Histological scores of inflammation (**E**) and cartilage damage (**F**). Values are the mean ± SEM. n.s., not significant. *, *p* < 0.05.

**Figure 5 ijms-21-04145-f005:**
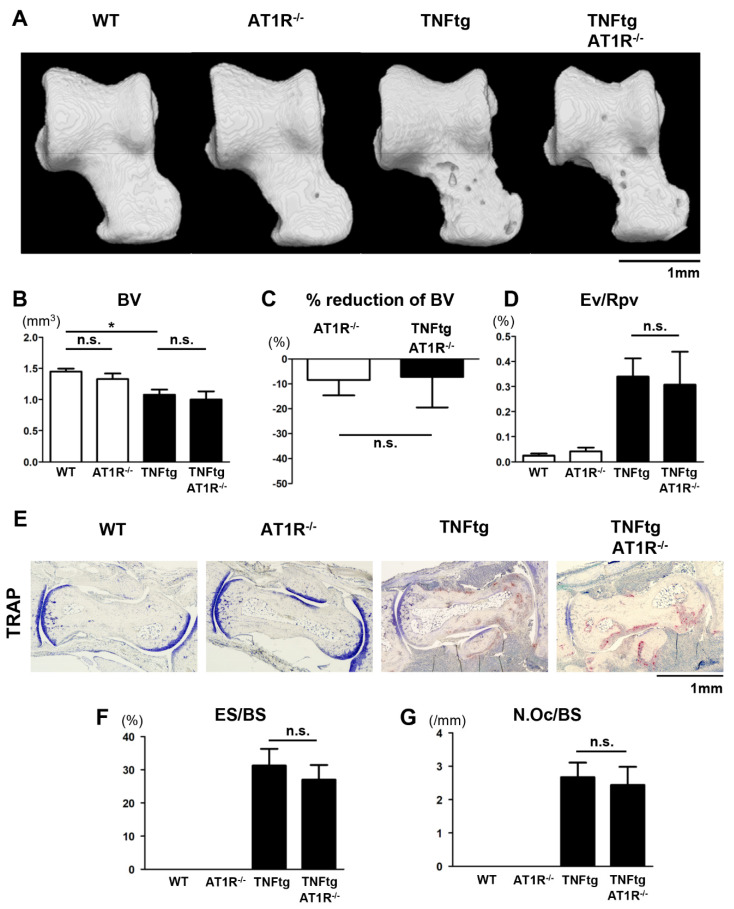
Influence of AT1R depletion on bone erosion in tumor necrosis factor-transgenic (TNFtg) mice. TNFtg mice were crossed with AT1R-deficient (AT1R^−/−^) mice. WT (*n* = 9), AT1R^−/−^ (*n* = 7), TNFtg (*n* = 12), and TNFtg AT1R^−/−^ (*n* = 8) male mice were analyzed at the age of 16 weeks. (**A**) Representative 3D micro-CT images of the talus. (**B**) Bone volume (BV) of the talus. (**C**) Rate of reduction in BV by AT1R deficiency relative to that in the control mice of each genotype. (**D**) Eroded volume per repaired volume (Ev/Rpv) of the talus. (**E**) Representative images of tartrate-resistant acid phosphatase (TRAP) staining around the talus bones of indicated mice; original magnification ×40. (**F**) Eroded surface per bone surface (ES/BS) around the taluses. (**G**) The number of osteoclasts per bone surface (N.Oc/BS) around the taluses. Values are the mean ± SEM. n.s., not significant. *, *p* < 0.05.

**Figure 6 ijms-21-04145-f006:**
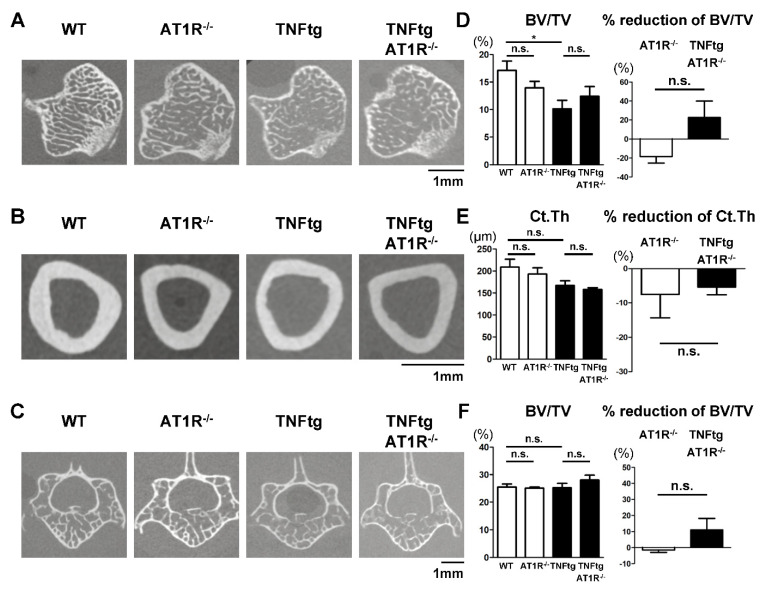
Effect of AT1R deficiency on bone properties of the trabecular and cortical bones in tumor necrosis factor-transgenic (TNFtg) mice. TNFtg mice were crossed with AT1R-deficient (AT1R^−/−^) mice. WT (*n* = 9), AT1R^−/−^ (*n* = 7), TNFtg (*n* = 12), and TNFtg AT1R^−/−^ (*n* = 8) male mice were analyzed at the age of 16 weeks. Representative 2D micro-CT images of the tibia trabecular bone (**A**), the tibia cortical bone (**B**), and the trabecular bone of the spine (fifth lumbar vertebra) (**C**). (**D**) Bone volume per total volume (BV/TV) and reduction rate of the tibia trabecula bone. (**E**) Cortical thickness (Ct.Th) and reduction rate of the tibia midshaft. (**F**) Bone volume per total volume (BV/TV) and reduction rate of the fifth lumbar vertebral trabecular bone. Values are the mean ± SEM. n.s., not significant. *, *p* < 0.05.

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
