# Peer review of "Effect of Angiotensin II on Bone Erosion and Systemic Bone Loss in Mice with Tumor Necrosis Factor-Mediated Arthritis"

_ijms, 2020, doi:10.3390/ijms21114145_

Round 1

Reviewer 1 Report

The paper is very well designed and performed. The results are sound

I suggest to cite the following paper which may be relevant for the clinical transfer of the data

Angiotensin II type 2 receptor (AT2R) as a novel modulator of inflammation in rheumatoid arthritis synovium Sci Rep. 2017;7(1):13293. doi: 10.1038/s41598-017-13746-w

Author Response

Dear Professor,

We greatly appreciate the comments from you and the reviewers regarding our manuscript entitled “Effect of angiotensin II on bone erosion and systemic bone loss in mice with tumor necrosis factor-mediated arthritis” (ijms-798392). We have carefully revised the manuscript according to the concerns raised by the reviewers, and specific points have been addressed below. We have used red font to denote all amendments to the text in the Word file.

Reviewer#1

The paper is very well designed and performed. The results are sound

I suggest to cite the following paper which may be relevant for the clinical transfer of the data

Angiotensin II type 2 receptor (AT2R) as a novel modulator of inflammation in rheumatoid arthritis synovium Sci Rep. 2017;7(1):13293. doi: 10.1038/s41598-017-13746-w

We appreciate the reviewer’s positive comments.

As the reviewer suggested, we discussed the possible involvement of AT2R in the process of bone erosion in arthritis. This description was added to the Discussion section of the revised manuscript at page 12, line 269.

Reviewer 2 Report

This manuscript use gain (Ang II administration) and lose (AT1R-/-) function approach to investigate the possible effect of the renin-angiotensin system (RAS) on the progression of joint erosion and systemic bone loss in rheumatoid arthritis using TNF-Tg arthritic model. The outcome is useful to understand the pathophysiology of RA. Some concerns need to be addressed.

  1. Fig.1. In addition to the mRNA expression, protein level of Agtr1 should also be tested.
  2. The rate of RA in female is 3 folds more than male. Why male TNF-Tg mice were used to evaluate the effect of AngII on the progress of RA?
  3. It is not clear how does Ang II administration exacerbate the joint erosion and increased osteoclast formation in the joints of TNF-Tg mice, directly or indirectly stimulating osteoclast formation, promoting angiogenesis? Quantification of osteoclasts in the joints of Fig.2 is missing.
  4. Fig.3. Ang II administration did not change trabecular bone mass in TNF-Tg mice. The trabecular bone mass did not have statistical difference between H2O treated WT and TNF-Tg mice. Was the statsictic analysis right? The image selected show trabecular reduction in TNF-tg mice compared to the WT mice and quantification also show this trend of reduction in the TNF-Tg mice. The number of mice (6 or 7) might not be enough? Have you done a power analysis?
  5. Fig.4. The joint erosion between TNFtgAT1R-/- mice and the control TNFtg mice did not have difference. The erosion of talus in the TNFtg mice was mild, shown by the images of micro-CT and histology compared to that shown in Fig.2. Have you chosen the right time point (mice age) ? 16-week-old TNFtg mice should have very serve joint erosion. Does the background of AT1R-/- mice affect the progression of RA in TNFtg mice?
  6. Which data support your claim that “Our findings suggest that, in patients in whom local AT1R is upregulated, joint  destruction can be promoted by activation of the systemic RAS”.

Author Response

Dear Professor,

We greatly appreciate the comments regarding our manuscript entitled “Effect of angiotensin II on bone erosion and systemic bone loss in mice with tumor necrosis factor-mediated arthritis” (ijms-798392). We have carefully revised the manuscript according to the concerns raised by the reviewers, and specific points have been addressed below. We have used red font to denote all amendments to the text in the Word file.

Reviewer#2

This manuscript use gain (Ang II administration) and lose (AT1R-/-) function approach to investigate the possible effect of the renin-angiotensin system (RAS) on the progression of joint erosion and systemic bone loss in rheumatoid arthritis using TNF-Tg arthritic model. The outcome is useful to understand the pathophysiology of RA. Some concerns need to be addressed.

We appreciate the reviewer’s positive comments. We have responded to the concerns as follows.

  1. Fig.1. In addition to the mRNA expression, protein level of Agtr1 should also be tested.

We agree that the validation of protein expression would support our findings obtained by gene expression analysis. We are continuing to explore the expression of RAS components in the inflammatory joints. However, because of the COVID-19 pandemic, researchers in our institute have been urged to minimize mouse colonies, and new studies are restricted. In this situation, the experiments including sampling will take substantial additional time to complete and should be considered for future studies. We would appreciate the reviewer’s understanding.

  1. The rate of RA in female is 3 folds more than male. Why male TNF-Tg mice were used to evaluate the effect of AngII on the progress of RA?

No significant sex differences in the severity of arthritis in the TNFtg mice have been reported, and no detectable sex differences have been detected based on our own observations. Regarding the reviewer concerns, bone properties of systemic bone can differ depending on sex. In our experiment, we analyzed only male mice, because female mice were needed to maintain the mouse colony. Since sex hormones differentially affect bone metabolism depending on sex, Ang II possibly exerts different effects on female TNFtg mice. The possible sex differences in the effect of Ang II will be clarified in future studies, but at this point, we believe that this issue is beyond the scope of the current paper. 

  1. It is not clear how does Ang II administration exacerbate the joint erosion and increased osteoclast formation in the joints of TNF-Tg mice, directly or indirectly stimulating osteoclast formation, promoting angiogenesis? Quantification of osteoclasts in the joints of Fig.2 is missing.

As the reviewer suggested, we have discussed the possible mechanisms through which Ang II exacerbates bone erosion with some additional experimental data. Based on this, Ang II did not directly induce osteoclast formation in bone marrow-derived macrophage cultures. These experimental data were included in Figure A4 of the revised manuscript. We also added the following description regarding the possible mechanisms in the Discussion section of the revised manuscript:

“The limitation of our study is that the precise mechanisms through which Ang II enhances bone erosion are unclear. We have tested the direct effect of Ang II on osteoclast differentiation in murine primary bone marrow-derived macrophage cultures but found no detectable promoting effect of Ang II on osteoclast formation (Figure A4). Ang II might dominantly regulate osteoclastic bone-resorbing activity or it could indirectly enhance osteoclastogenesis via synovial cells, osteoblasts, or osteocytes. Other possibilities are that Ang II regulates angiogenesis in arthritic joints or that Ang II modulates cellular functions via Ang II type 2 receptor, which reportedly regulates inflammation in the arthritic synovium [21]. Further research will be required to clarify the underlying mechanisms.”

We additionally performed a quantitative analysis of osteoclast formation. We quantified the eroded surface per bone surface (ES/BS) and the number of osteoclasts per bone surface (N.Oc/BS) around taluses. The quantitative data are included in the revised manuscript as Figure 2F and 2G. In addition, we also performed quantitative analyses for Figure 5. The data were also included in the revised manuscript as Figure 5F and 5G.

  1. Fig.3. Ang II administration did not change trabecular bone mass in TNF-Tg mice. The trabecular bone mass did not have statistical difference between H2O treated WT and TNF-Tg mice. Was the statsictic analysis right? The image selected show trabecular reduction in TNF-tg mice compared to the WT mice and quantification also show this trend of reduction in the TNF-Tg mice. The number of mice (6 or 7) might not be enough? Have you done a power analysis?

In Figure 3D, H2O-infused TNFtg mice exhibited decreased trabecular bone mass compared to that in H2O-infused WT mice, but the difference did not reach statistical significance. As the reviewer suggested, the number of mice in the experiment, especially H2O-infused TNFtg mice, might have been too small. We used littermate controls to set the experimental groups. The number of H2O-infused TNFtg mice was unintentionally small at that time. As per the reviewer’s concern, we speculate that the relatively small sample size of the H2O-infused TNFtg mouse group masked the differences in bone loss between H2O-infused TNFtg and WT mice. Indeed, when we analyzed more than nine mice (WT, n = 9; TNFtg, n = 12) in Figure 6D, statistically significant differences in the bone loss were observed between WT and TNFtg mice.

We did not perform power analysis in advance of this study. We would like to perform this analysis in future experiments to set sample sizes appropriately.

  1. Fig.4. The joint erosion between TNFtgAT1R-/- mice and the control TNFtg mice did not have difference. The erosion of talus in the TNFtg mice was mild, shown by the images of micro-CT and histology compared to that shown in Fig.2. Have you chosen the right time point (mice age) ? 16-week-old TNFtg mice should have very serve joint erosion. Does the background of AT1R-/- mice affect the progression of RA in TNFtg mice?

We have not tested different time points in this series of experiments. The TNFtg mice had already exhibited severe arthritis at the age of 16 weeks. Further, the osmotic pumps used in this study can function for a maximum of 4 weeks. That is why we set the experimental period from 12 weeks (mild arthritis) to 16 weeks (severe arthritis) of age. As the reviewer suggested, arthritis in the TNFtg mice can progress with age, even after 16 weeks of age. Thus, the effect of Ang II or AT1R deficiency might be different depending on the age. This point should be clarified in future studies.

Both TNFtg and AT1R−/− mice have a C57BL/6 background. However, as the reviewer pointed out, the original strains of TNFtg mice and AT1R−/− mice are C57BL/6N and C57BL/6J backgrounds, respectively. Since there are some genetic differences between C57BL/6N and C57BL/6J, the differences might have slightly affected the severity of arthritis in our mouse studies.

  1. Which data support your claim that “Our findings suggest that, in patients in whom local AT1R is upregulated, joint destruction can be promoted by activation of the systemic RAS”.

We agree with the reviewer’s comment. Since the upregulation of AT1R expression in joints has been reported only in arthritic conditions, our statement in the original manuscript could be misleading. We intended to state that the effect of increased Ang II in joints can be mediated via increased imported Ang II from circulation in some pathological conditions.

To clearly state this point, we replaced our statement as follows:

“Our findings suggest that in patients in whom the local effect of Ang II is upregulated via increased imported Ang II from circulation, joint destruction can be promoted as a consequence of systemic RAS activation.”

Round 2

Reviewer 2 Report

The answer to the questions raised are unacceptable.

I understand that new animal studies are restricted due to the COVID-19 pandemic. However, the required experiments for article and grant submission would be possible, in particular, the samples collected previously for some data reported in this manuscript can be used to test the protein level of Agtr1 by immunostaining.

The sex difference of the severity of arthritis in TNF-Tg mice has been published (Bell RD, et al. Arthritis Rheumatol. 2019 Sep;71(9):1512-1523).

Enough number of mice, determined through power analysis, is required for this manuscript and for any publication to achieve conclusion for the findings.

The authors should provide a convincing evidence to demonstrate why the joint erosion in TNF-Tg mice has so big difference between Fig.4 and Fig.2.

Author Response

Dear Professor,

We appreciate the reviewer’s comments to improve our manuscript. Following the reviewer’s comments, we performed some additional experiments and revised our manuscript with some new data.

I understand that new animal studies are restricted due to the COVID-19 pandemic. However, the required experiments for article and grant submission would be possible, in particular, the samples collected previously for some data reported in this manuscript can be used to test the protein level of Agtr1 by immunostaining.

We appreciate the reviewer’s understanding of our situation. Fortunately, a state of emergency in this area had been lifted on May 15. Afterward, we performed additional experiments requested by the reviewer.

We performed western blot analysis and immunohistostaining to test the protein levels of angiotensin II type 1 receptor (AT1R). The data reveal that AT1R protein expression is increased in arthritic joints, which is in accordance with the increased mRNA levels. We have included these new results in Figure A1.

The sex difference of the severity of arthritis in TNF-Tg mice has been published (Bell RD, et al. Arthritis Rheumatol. 2019 Sep;71(9):1512-1523).

We appreciate the reviewer’s comment. The paper analyzed the phenotypes of Tg3647, a strain of TNFtg mice. The paper reported that female Tg3647 mice develop earlier and more severe arthritis than male Tg3647 mice. The findings reported in the paper are beneficial to better understand the differential phenotypes of TNFtg mice depending on strains. Currently, several strains of TNFtg mice are available, and some phenotypical differences have been reported (Springer Semin Immunopathol. 2003 Aug;25(1):19-33.). The phenotypical differences are thought to attribute to several factors, such as the designs of the targeting vectors, copy numbers of the transgenes, and transcriptional regulatory regions surrounding the randomly inserted transgenes (Springer Semin Immunopathol. 2003 Aug;25(1):19-33, The EMBO Journal. 1991, 10(13):4025-4031.).

The TNFtg mice used in our study were generated by Taconic Biosciences, which is a relatively newly established TNFtg strain. In terms of the sex difference in arthritis, female TNFtg mice of Taconic have been reported to exhibit milder phenotypes (BMC Physiology 2007, 7(13); 1-16), although the tangible data were not presented. The milder phenotype of the female TNFtg mice (Taconic) is contrary to those of Tg3647 mice, suggesting the presence of some genetic differences between the two TNFtg strains. As we described in the earlier Response Letter, no detectable sex differences have been noticed based on our own observations. Thus, we assume that female TNFtg mice established by Taconic exhibit comparable or, if there is, milder levels of arthritis compared to male TNFtg mice.

The next question is whether the effects of Ang II and AT1R deficiency would vary depending on TNFtg strains. This point would be important to generalize our findings obtained in this study. Thus, we would like to test this point using a different TNFtg strain or other arthritis models in our future study.

Fig.3. Ang II administration did not change trabecular bone mass in TNF-Tg mice. The trabecular bone mass did not have statistical difference between H2O treated WT and TNF-Tg mice. Was the statsictic analysis right? The image selected show trabecular reduction in TNF-tg mice compared to the WT mice and quantification also show this trend of reduction in the TNF-Tg mice. The number of mice (6 or 7) might not be enough? Have you done a power analysis?

Enough number of mice, determined through power analysis, is required for this manuscript and for any publication to achieve conclusion for the findings.

We apologize that we had misunderstood the reviewer’s comment. We initially thought that the reviewer referred to a priori power analysis that we should have done before our experiments and advised us to do the prior analysis for our future experiments. Now, we realized that the reviewer had suggested to perform post-hoc power analysis. Thus, we have done the post-hoc power analysis following the reviewer's suggestion.

Post-hoc power analysis was performed to determine the number of mice that would have been required to detect statistically significant differences in the BV/TV of trabecular bone of tibia in Figure 3D. We analyzed the data using SigmaPlot v14.0 (Systat Software Inc.), and the following parameters were applied: desired power = 0.80, 2 tails, and alpha = 0.05 for ANOVA. The power analysis shows that at least 15 mice would be required for each group to find a significant difference. We speculate this is due to a large variation of the data in H2O- and Ang II-treated WT groups. Thus, this set of experiments with small sample size may have insufficient statistical power to detect a substantial difference. Indeed, in a similar comparison shown in Figure 6D, we used 9 WT mice and 12 TNFtg mice and found a statistically significant difference. Therefore, we consider that there is a true difference in the trabecular BV/TV between WT and TNFtg mice, but the numbers we used for Figure 3D were insufficient to demonstrate the statistically significant difference, as the reviewer pointed out. This possible limitation of our study was mentioned on Page 12 in the Discussion section.

The authors should provide a convincing evidence to demonstrate why the joint erosion in TNF-Tg mice has so big difference between Fig.4 and Fig.2.

We pondered over the question that the reviewer had asked. In the earlier Response Letter, we addressed that there were the differences in the background of the mice; TNFtg mice are on C57BL/6N, and AT1R-/- strain mice are on C57BL/6J. In addition to that, whether osmotic pumps were implanted or not made another difference between the H2O-treated TNFtg mice and TNFtg mice on the AT1R-/- strain.

In terms of the phenotypical differences, we compared the quantitative data between the H2O-treated TNFtg mice and TNFtg mice on the AT1R-/- strain. Data were presented in a table below. The analyses revealed no statistically significant differences, although the sample size of H2O-treated TNFtg mice is small. The results suggest that background differences or implantation of the osmotic pumps have limited or negligible influence on the bone erosion of the TNFtg mice.

H2O-treated

TNFtg

(n = 4)

TNFtg

(AT1R-/- strain)

(n = 12)

p-value

Inflammation (0–4)

3.0 ± 0.2

3.1 ± 0.3

0.797

Cartilage damage (0–4)

3.9 ± 0.1

3.5 ± 0.2

0.323

BV (mm3)

1.16 ± 0.05

1.08 ± 0.08

0.607

Ev/Rpv (%)

0.23 ± 0.05

0.34 ± 0.07

0.405

ES/BS (%)

29.4 ± 5.6

37.5 ± 3.3

0.219

Values are means ± SEM. P-values were determined by student t-test.

              In the revised manuscript, we added new data relating to the possible mechanisms of Ang II-mediated bone erosion, since the reviewer expressed the concern about the mechanisms. Since this issue remains unresolved, we undertook experiments to determine whether Ang II could indirectly increase osteoclast formation via other cells. Bone marrow-derived macrophages and neonatal calvarial osteoblasts are co-cultured in the presence of Ang II. In the co-culture, Ang II enhanced osteoclast formation, suggesting that Ang II enhances osteoclast formation indirectly through stromal cells. These findings are consistent with previous findings (J Bone Miner Res. 2009.24(3), 241-250). We included the data of the co-culture as Figure A5 and described the possible mechanisms in the Discussion section of the revised manuscript.